# A Theoretical Perspective on Why Socioeconomic Health Inequalities Are Persistent: Building the Case for an Effective Approach

**DOI:** 10.3390/ijerph19148384

**Published:** 2022-07-08

**Authors:** Lisa Wilderink, Ingrid Bakker, Albertine J. Schuit, Jacob C. Seidell, Ioana A. Pop, Carry M. Renders

**Affiliations:** 1Department of Health Sciences, Faculty of Sciences, Amsterdam Public Health Research Institute, Vrije Universiteit Amsterdam, 1081 HV Amsterdam, The Netherlands; j.c.seidell@vu.nl (J.C.S.); carry.renders@vu.nl (C.M.R.); 2Department of Healthy Society, Windesheim University of Applied Sciences, 8017 CA Zwolle, The Netherlands; i.bakker@windesheim.nl; 3School of Social and Behavioral Sciences, Tilburg University, 5037 AB Tilburg, The Netherlands; jantine.schuit@tilburguniversity.edu (A.J.S.); i.a.pop@tilburguniversity.edu (I.A.P.)

**Keywords:** socioeconomic health inequalities, narrative literature review, theoretical models, systems thinking, complex whole-system approach

## Abstract

Despite policy intentions and many interventions aimed at reducing socioeconomic health inequalities in recent decades in the Netherlands and other affluent countries, these inequalities have not been reduced. Based on a narrative literature review, this paper aims to increase insight into why socioeconomic health inequalities are so persistent and build a way forward for improved approaches from a theoretical perspective. Firstly, we present relevant theories focusing on individual determinants of health-related behaviors. Thereafter, we present theories that take into account determinants of the individual level and the environmental level. Lastly, we show the complexity of the system of individual determinants, environmental determinants and behavior change for low socioeconomic position (SEP) groups and describe the next steps in developing and evaluating future effective approaches. These steps include systems thinking, a complex whole-system approach and participation of all stakeholders in system change.

## 1. Introduction

Socioeconomic health inequalities are a challenge for health policymakers and civil society. In many affluent parts of the world, including the Netherlands, people with a relatively low socioeconomic position (SEP) live shorter lives in good health than people with a high SEP [1,2]. Despite policy intentions and many interventions in recent decades in the Netherlands, these health inequalities have not been reduced [3].

Differences in health-related behaviors such as smoking, alcohol consumption, dietary patterns and physical activity, contribute to socioeconomic health inequalities [4]. Those inequalities can be caused by numerous determinants at the individual level that influence health-related behaviors and are unequally distributed across different SEP groups, in favor of high-SEP groups. Examples of such determinants include chronic mental stress, lower self-efficacy for behavioral change, a lack of knowledge about healthy choices and a lower level of health literacy [5,6,7,8]. Moreover, a wide range of determinants at the population level influence health-related behaviors. It is known that characteristics of the (socio)economic, political, cultural and physical environment (micro, meso and macro) play a role, since an unfavorable environment often offers less support and fewer opportunities for healthy behavior and thus contributes to health inequalities [9,10,11,12]. For example, low-SEP neighborhoods are usually characterized by less public recreation space to be physically active, more marketing of unhealthy foods and drinks and a higher density of fast-food restaurants [13,14]. Moreover, the social environment plays a role, since people with a relatively low SEP often experience less social support from peers and family to change their (unhealthy) behavior (e.g., prevailing norms and example behavior) [15,16]. Furthermore, a systematic review of neighborhoods and health shows that poorer safety, more violence and less social cohesion in disadvantaged neighborhoods negatively affect healthy behaviors such as being physically active [17]. For example, it is known that parents hesitate to let their children play alone in public playgrounds if they experience social unsafety and a lack of social control, which is more often the case in low-SEP neighborhoods [18].

A meta-analysis on the effectiveness of health-related behavior interventions showed that interventions are less effective for low-SEP groups in comparison to the general population [19]. This suggests that interventions do not sufficiently reach low-SEP groups and may not be well tailored to the needs of this particular target group. Moreover, a study on equity-specific effects of health-related behavior interventions showed a lack of evidence in the literature for developing interventions that are well suited to the needs and opportunities of people from these low-SEP groups [20]. Equity-specific effects mean that behavioral interventions may have more impact in higher SEP groups, for example, because of differences in participation levels between low and high SEP groups. Moreover, the cognitive capacity hypothesis implies that interventions focusing on generating more knowledge about healthy behavior contributes to widening health inequalities because these interventions are more in line with higher SEP groups’ cognitive capabilities [20]. We know that in order to tackle inequalities, it is important to take into account determinants at the individual as well as the different environmental levels and pay attention to their coherence and synergy [21,22,23,24]. Multiple complex interventions and integrated approaches that acknowledge this have been developed in recent years but have not yet succeeded in properly aligning these interventions to the living environment, living conditions, skills, wishes and needs of people from low-SEP groups [11].

Several theoretical models stress the importance of taking into account both the individual level and the different environmental levels to promote healthy behavior and reduce socioeconomic health inequalities. Bronfenbrenner’s Ecological Systems Theory, for example, organizes different levels of external influence on individual behavior, from most proximal to distal: micro, meso, exo and macro [25,26]. Dahlgren and Whitehead’s Model of Health Determinants (see Figure 1 [27]) also has a widespread impact in research on health inequality and displays different levels of influence ranging from individual lifestyle factors to the level of social and community networks and general socioeconomic, cultural and environmental conditions. Both models acknowledge the wide range of determinants from different levels that influence behavior.

This rainbow model remains one of the most effective illustrations of health determinants. However, sometimes this model has been criticized for not sufficiently highlighting the interaction between different determinants. A growing body of literature shows that individuals do not make (un)healthy behavior choices based on their individual intention only, but that they are influenced by opportunities and restrictions in their socio-economic, cultural and physical environment [28,29,30,31]. Individuals can respond to their environment in two ways: rationally or automatically, as behaving in a certain way does not always happen consciously. Socioeconomic health inequalities can be the result of the interaction of a wide range of determinants at the individual and the environmental level. The socio-ecological models of Bronfenbrenner, and of Whithead and Dahlgren, assume that determinants interact at different levels but do not show how [32,33]. The nature and direction of the interrelationships of determinants often remain elusive.

Another reason why current models have not been able to contribute to the reduction of socioeconomic health inequalities is that they do not sufficiently connect to the living environment, living conditions, skills, wishes and needs of specific population groups. Dahlgren and Whitehead’s model, for example, includes generic determinants for the population as a whole, while those determinants that influence health-related behaviors may do so differently for low-SEP groups [34]. Due to a different socioeconomic context, different mechanisms may influence the relationships between determinants of different levels in this system. For example, exposure to an unfavorable environment may result in poorer health outcomes for someone with a low SEP compared with someone with a higher SEP, because they are more vulnerable to the consequences of that environment due to insufficient capabilities and opportunities (e.g., resources, skills, social connections and power) to respond to this environment.

Not fully understanding how the dynamics of determinants of health-related behaviors operate for people from low-SEP groups means that it is not entirely clear how health inequalities can be reduced. In addition, since we are not succeeding in reducing socioeconomic health inequalities, it appears we have insufficient knowledge on how interventions and approaches can contribute to a reduction. Which combinations of contextual factors and mechanisms can lead to the outcome of reduced health inequalities? A deeper understanding from a theoretical perspective is needed to investigate behavioral change, in particular, in relation to reducing socioeconomic health inequalities. A study by Øversveen and colleagues on health inequalities argues in favor of ‘the use of sociological theory to develop more dynamic models that enhance the understanding of the complex pathways and mechanisms linking social structures to health’ [35]. To date, such use appears to be lacking.

This paper aims to increase insight into why socioeconomic health inequalities are so persistent and build a way forward to improved approaches from a theoretical perspective. A narrative literature review is done to analyze and summarize relevant theories [36]. A narrative review is a discussion of important topics from a theoretical point of view, with a less formal approach than a systematic review. This type of review does not include the reporting of search terms, inclusion and exclusion criteria and used databases. Given this, firstly, we will discuss relevant behavioral theories that focus solely on the individual level. Following narrative literature review principles, distinctive theories were selected that are commonly used and cited when understanding human behavior change. Thereafter, we will discuss theories that acknowledge interactions between individuals and their environment. Lastly, we will show the complexity of the system of individual determinants, environmental determinants and behavior change for people from low-SEP groups and describe the next steps in developing, implementing and evaluating future improved approaches.

## 2. Theories Explaining Health-Related Behaviors Focusing Solely on the Individual Level

Before the emergence of modern medicine after the Industrial Revolution in the 18th century and the widespread process of medicalization ever since, public health was mainly about preventive measures focused on the community level [37]. For example, preventing plagues and other infectious diseases was mainly about improving living conditions and hygiene. In the past century however, many theories were developed to explain behavior change focusing solely on the individual level and coming from biology, medicine and psychology. This started with Pavlov (1927), whose theory on classical conditioning showed that creating or changing personal associations can be used to influence behavior [38]. Subsequently, other scientists developed theories explaining behavior change. We describe three commonly used and criticized models: the theory of planned behavior, the Health Belief Model and the transtheoretical model.

### 2.1. The Theory of Planned Behaviour

A cognitive-oriented model that fully focuses on individual determinants is the theory of planned behavior, first described by Ajzen in 1985 [39]. The theory implies that behavior can be predicted from attitudes toward the behavior, subjective norms and perceived behavioral control (see Figure 2 [39]). This suggests that a person’s health behavior is determined by the intention they have to perform this behavior. This theory is, for example, used in a school-based intervention program to promote physical activity [40].

### 2.2. The Health Belief Model

Rosenstock’s health belief model (1974) deals with health behavior change in particular. It also focuses only on individual cognitive determinants and describes five concepts that can predict why someone would take action to prevent illness conditions: susceptibility; severity; benefits and barriers to a behavior; cues to action; and self-efficacy (see Figure 3 [41]). In short, it implies that people are more likely to behave in a certain way (to reduce risks) if they believe that they will be affected by the illness condition and that this will have consequences [42]. Moreover, the health belief model implies that people should believe that adopting different behaviors would help reduce the risk, would not take too much effort in terms of barriers or costs, and that people are able to change the behavior. An example of how this model is used can be found in a study on a nutrition education worksite intervention for university staff [43].

### 2.3. The Transtheoretical Model/Stages of Change Model

Prochaska & Di Clemente (1982), authors of the transtheoretical model/stages of change model, state that no single theory can describe the complexity of behavior change and, therefore, integrated principles of leading theories of behavior change [44]. The transtheoretical model is based on the idea that behavior change does not happen in one single moment but across five different stages of change: pre-contemplation, contemplation, preparation, action and maintenance (see Figure 4 [44]). Nevertheless, the model focuses only on the individual. This theory is, for example, used in a motivational nursing intervention with heart failure patients [45].

### 2.4. Limitations of Theories That Focus Solely on the Individual Level

Although we learned from theories focusing solely on the individual that characteristics such as motivation and skills are important when influencing health-related behaviors, we should mention certain criticisms. The main criticism of these theories is that they pay no attention to the socio-cultural and physical environment in which these behaviors occur [23].

The theory of planned behavior, for example, does not take opportunities and resources into account. If a person intends to perform a particular behavior, but the environment does not provide the opportunity for or even impedes the behavior, it is less likely to happen. Emotions or affective influences are not taken into account in the model either, although we know that they greatly influence performing specific behaviors as well [46,47]. An important limitation of the health belief model is that it does not take a person’s attitudes and beliefs into account, pays no attention to habits and does not take environmental/economic influences into account. Moreover, it assumes that every person has access to the same level of information about the illness or disease, while we know that health literacy is a common issue among people from low-SEP groups [48]. An often-mentioned limitation of the transtheoretical model is that it assumes that individuals make logical, rational plans. Overall, theories that focus solely on the individual level emphasize the cognitive individual, with no attention to non-rational influences, the environment and interactions with this environment.

## 3. Theories Explaining Health-Related Behaviors That Include the Individual and Environmental Level and Their Interactions

Multiple theories that take into account both the individual and environmental levels help explain healthy behavior since we know that determinants from both levels play a role and interact. We describe seven commonly used theories that acknowledge the interaction between the individual and the environmental level: social networks and support theory, social capital theory, the capability approach, structuration theory, social cognitive theory, cultural capital theory and the default lifestyle theory/fundamental causes theory. Our aim is not to provide a complete overview of the relevant theories but to focus on those theories that we believe are relevant to understanding behavior change in low-SEP groups.

### 3.1. Social Capital Theory/Social Networks and Support Theory

There is no unified theory that describes the relationship between social networks and support and health. However, it has long been known in the literature that some of the most powerful influences on a person’s behavior are attitudes and resources within a person’s social network [49,50]. Figure 5 shows how social capital relates to self-related health [49]. Research shows that social network structure, or even the perceived availability of supportive functions, is related to lower levels of performing unhealthy behaviors [51].

Studies into social capital and health are often traced back to the work of Durkheim (1897), who showed that social integration was inversely related to the suicide rate in societies. Social capital is still an important social determinant of physical and mental health and can be defined as being part of social groups that yield reproductive benefits [52,53]. Social capital can provide resources, support, influence and obligations for specific behavior. Social relations can, for example, be supportive of showing healthy behavior, and it is known that people are healthier when people close to them care about them in the event of illness [9]. In addition, strong relational ties can result in stress reduction. Social capital can be measured by the quantity and quality of social networks and personal relationships that a person has. Social support theory is, for example, used in a physical activity behavior intervention with children [54].

### 3.2. Structuration Theory

Anthony Giddens’ structuration theory (1984) explains the relationship between the individual and their environment in the reproduction of inequalities [55]. The theory from sociology proposes a bidirectional relationship between structure (environment) and agency (individual), where both can function as the cause and effect of the other (see Figure 6 [56]). Social structures create constraints and opportunities for particular behavior. At the same time, individuals are agents whose behavior creates and transforms social structures. In other words, environments nor individuals dominate in the structuration of society. They cannot be conceived independently. For example, cafeterias located next to schools provide opportunities for pupils to snack, but it is the purchasing behavior of the children that makes cafeteria owners want to have their business next to the school. Giddens defines social structures in terms of rules and resources. Examples of resources include physical features of the environment or the capacity to organize life chances. Rules refer to the learned procedures and techniques that are needed for performing the behavior in relation to structural constraints and opportunities. Bernard et al. [57] conceptualize neighborhoods as providers of resources related to health-related behaviors and the production of health inequalities. An example of how the structuration theory can be used in practice can be found in a study on the interplay of structure and agency in health promotion [58].

### 3.3. Social Cognitive Theory (SCT)

The social cognitive theory, first described by psychologist Bandura (1989) is rather similar to the structuration theory. Like Giddens, it states that human behavior is the product of the dynamic interplay of personal, behavioral and environmental influences (see Figure 7 [59]). Social cognitive theory is, for example, used in a nutrition and physical activity intervention among web-health users [60]. Although it notes how environments can shape behavior, this theory mainly focuses on people’s abilities to adjust and create environments to suit the purposes they create for themselves. In addition to a person’s individual capacity to interact with their environment, this theory emphasizes the human capacity for collective action, enabling individuals to work together in organizations and social systems to achieve changes in the environment that benefit the entire group.

### 3.4. Cultural Capital Theory

The concept of cultural capital was first described by Bourdieu in 1986 [61]. The concept is about the social resources of a person that promote social mobility in a stratified society and can be seen as a set of habits, practices and preferences with which the higher SEP groups in particular distinguish themselves from lower SEP groups (see Figure 8). Because cultural capital is intergenerationally transferable, cultural capital theory is useful in understanding the persistence of inequalities between generations. The British sociologist Goldthorpe [62] analysed inequality in terms of social relations in the context of which individuals are, in some sense, advantaged or disadvantaged. In his view, social stratification is inequality of a structured, social kind. According to Goldthorpe, the positions that individuals hold in social stratification will determine their life chances and lifestyles. Moreover, a recent study into cultural capital in relation to food choices showed that higher educated people possess more cultural capital and that those with a high level of cultural capital make healthier food choices than people with a lower cultural capital [63]. An example of how cultural capital theory can be applied is shown in the study of Shim et al. [64].

### 3.5. The Default Lifestyle Theory, Fundamental Causes Theory

Sociologists Mirowsky and Ross (2015) state that too much food and insufficient physical activity, or what they refer to as ‘the default lifestyle’, is the standard model of life today [65]. Default means that an option is taken automatically; it is what happens unless you actively reject it. In their paper, Mirowsky and Ross mention three trends that discredit the body systems that stimulate health. The first is about displacing human energy with mechanical energy, which, according to the authors, leads to less physical activity in daily life. The second is about displacing household food production with industrial food production, which leads to dietary excesses. The third is about displacing health maintenance with medical dependency.

Mirowsky and Ross conclude that education helps individuals recognize the health risks of the default unhealthy lifestyle. They even indicate that education is the key to socioeconomic differences in health because controlling one’s own life is harder for the least advantaged. Less education means less development of the abilities needed to reject the default lifestyle, e.g., knowledge, critical analysis, long-range strategic thinking, personal agency and self-direction.

The theory of fundamental causes has the same analogy as the default lifestyle theory on the capacity to reject an unhealthy lifestyle. It was first described by Link and Phelan in 1995 [66,67] and later by Mackenbach [68]. According to the theory of fundamental causes, social forces underlying social stratification cause health inequalities, and not exposure to the proximal risk factors, which are usually studied. According to this theory, health inequalities are the result of a person’s socioeconomic status providing them with so-called “flexible resources”. Examples of such resources include knowledge, money, power, prestige and social connections that can help avoid unhealthy risks or cope with the consequences of illness. A study by Phelan and Link [66] showed how social conditions are related to socioeconomic inequalities of health.

### 3.6. Limitations of Theories Explaining Health-Related Behaviours That Include Interaction of the Individual and Environmental Level

Theories that take determinants of the individual and environmental level and their interaction into account are more appropriate for understanding health inequalities. Therefore, these theories are a step in the right direction. They can help explain socioeconomic health inequalities since they indicate interactions in these systems that explain why behavior change can be different for people from low-SEP groups. For example, how a person’s social network can support healthy behavior, particularly for an individual who is struggling to find motivation, or how the environment can provide opportunities for healthy behavior.

However, these theories do not explain why socioeconomic health inequalities are not reduced or what future effective interventions and approaches should look like. Moreover, some specific criticism of the described theories should be mentioned. The theories show that interactions are present but do not inform us of how the relationship between a person, the environment and behavior works. In addition, the social cognitive theory assumes that changes in the environment will automatically lead to changes in individual behavior, although this is not always the case. The interplay between individual, behavior and environment is acknowledged, but the extent of each of these three is not clear. Another limitation of the default lifestyle theory is that it suggests that education is key because education helps to recognize the default unhealthy options. However, despite numerous implemented health education interventions, this has not led to a reduction in health inequalities [69]. This is probably due to the fact that education alone is not enough to overcome the overriding effects of unfavorable socioeconomic, cultural, and physical environmental determinants of behavior.

The main criticism of the described theories is that they do not show the full complexity of how determinants and their interactions contribute to the origin and persistence of socioeconomic health inequalities. In recent years, behavioral science has increasingly acknowledged that human behavior can be seen as a nonlinear and complex system. Nonlinear implies that the output of the system is not proportional to the input. For behavior change, this implies that the same input (e.g., improving a mix of individual and environmental determinants) does not always have the same output (behavior change) as a consequence [70]. This also means that an intervention aimed at changing one or more determinants does not have the same consequence in behavior change for different persons, or the same person at different moments in time. Moreover, the system can be seen as complex due to the variability in the extent to which changing determinants will lead to behavior change and the interactions between the different determinants that also affect the behavior change output. Systems thinking is needed to deal with this nonlinear complexity in promoting health behavior in low-SEP groups [29].

## 4. The Complexity of Changing Health-Related Behaviors in Low-SEP Groups

In this third part, we acknowledge the nonlinear complexity of promoting health behavior in low-SEP groups. We have shown in the previous paragraph that it is important to take into account the interaction between the various determinants of the individual level and the other levels, as described in the Dahlgren and the Whitehead and Bronfenbrenner model. On this basis, we can conclude that health inequalities have not yet been reduced because the complexity of the system of determinants from multiple levels and their dynamic interactions is not sufficiently acknowledged in current research, policy and practice. We describe how systems thinking, a whole system approach and participation of all stakeholders are the next steps in developing and evaluating effective approaches aimed at reducing socioeconomic health inequalities.

### 4.1. Systems Thinking: The Complex System of the Person, Ball and Slope

Theories that focus on the individual are based on the idea that determinants such as knowledge, attitudes, skills, competencies and self-efficacy can contribute to behavior change. However, as shown in theories that focus on the interaction of the individual and the environment, the opportunities and restraints in the environment are also of great influence. Therefore, it is important to look at the entire system in which behavior change takes place. This system of behavior change can be visualized as seen in Figure 9, showing the individual (person), the environment (slope) and the desired behavior change (the ball). All three can vary since certain individual determinants of health-related behaviors can vary: the person can be visualized as ‘stronger’ when the individual determinants positively contribute to behavior change or as ‘weaker’ when the individual determinants affect the behavior change less positively. The environmental determinants may vary as well: the slope can be steep if the environmental determinants influence health-related behaviors less positively or less steep if the environmental determinants influence health-related behaviors more positively. Lastly, the ball that needs to be pushed forward visualizes the desired behavior change, which can be smaller if the desired behavior change is relatively minor or bigger if the desired change is relatively major. Following this visualization, interventions aimed at behavior change for people from low-SEP groups who have to push a relatively big ball can be aimed at making the person stronger, the slope less steep and the ball smaller.

While this visualization can assist in our thinking, it does not completely visualize the complexity. In Figure 9, the individual, environment and desired behavior change are shown as three independent components within the system. The interactions between the aspects within the system and the mechanisms that influence the system for low-SEP groups are missing. In other words, the complex interaction between the individual and the environmental level is not shown. For example, compared with a person with a high SEP who has the skills and capabilities to push the ball forward, this might feel more challenging than for a person with a low SEP because they might have fewer skills and capabilities to react to the already steep slope. For example, if the person has limited health literacy [71], has had fewer opportunities in education or experiences (chronic) stress due to, e.g., relational problems or poverty. The authors of the scarcity theory [72] state that scarcity (for example, the scarcity of money) leads to chronic stress and causes a shortage of brainpower and capability to process information, and this shortage, in turn, contributes to scarcity. Experiencing scarcity can impede being able to pay attention, make good decisions, hold on to plans and resist (unhealthy) temptations [73]. Dynamic feedback loops of this type which can hinder or support the effects of actions and interventions are not shown in the figure with the person, ball and slope. Figure 9 is partly in line with the conceptual framework of the ‘behavior change ball’ developed by Hendriks et al. [74]. It features a visualized mountainous landscape in which the ‘behavior change ball’ has to roll. In their metaphor, too, the context can be steep or less steep to show how much the context contributes to (or hinders) change.

Interactions between the individual and the desired behavior change are relevant in reducing health inequalities as well. It is possible, for example, to help the individual experience the ball as less heavy, resulting in behavior change being started and persevered at more easily. This can be done, for example, by applying social marketing strategies. Social marketing is defined as ‘the use of marketing principles to influence a target audience to voluntarily accept, reject, modify or abandon behavior for the benefit of individuals, groups, or society as a whole’ [75]. Part of social marketing is to generate insights into what people from a defined target group, for example, people from low-SEP groups, need to start behaving differently and finding out what motivates and activates them. Interventions and activities can then be developed based on these insights that are linked to what these people from low-SEP groups find useful or attractive to do. For example, a physical activity such as walking could be linked to a social activity because the target group of older adults indicated that they would like to have more social contact with other older adults in the neighborhoods. These activities and interventions may require fewer individual capabilities and skills if they match their preferences. Following the principles of social marketing, the ‘costs’ experienced (e.g., money, time, effort, social consequences) of behaving differently should be lower than the value that it is perceived to bring. Nudging (providing ‘easy’ indirect suggestions as ways to influence behavior and change automatic behavior) is a strategy that can help people perceive the ‘costs’ as lower [76]. Moreover, the ball can also be made smaller by dividing big behavior change goals into smaller steps.

While interactions between the individual, the environment and the desired behavioral change are crucial in understanding the reasoning behind the origin and persistence of socioeconomic health inequalities, much is still unknown. We do know that the individual can be supported in recognizing, creating and taking advantage of (created) opportunities in the environment. We know that the social environment, for example, the individual’s social networks, can be more or less supportive in recognizing, creating and taking advantage of those opportunities. Conversely, we also know that making the environment more advantageous makes the individual’s healthy behavior more likely to happen. Moreover, social marketing techniques can make the desired behavior change be perceived as easier. Providing insights into those interactions and how they work for people from low-SEP groups specifically can help reduce health inequalities.

In recent years, it has become evident that to understand socioeconomic health inequalities, systems thinking is needed in which we not only identify the determinants from the different levels but also the interaction between determinants at different levels [77,78,79]. In other words, we need to consider the ‘bigger picture’. Several aspects of systems thinking differ from traditional thinking. For example, a traditional way of considering approaches and interventions is to develop and evaluate linear causes and effects, whereas in complex systems thinking, dynamic feedback loops are taken into account. Moreover, traditional ways of developing and evaluating approaches and interventions focus on individual, isolated activities, whereas in complex systems thinking one examines how systems work as a whole [29,80]. Complex systems thinking involves multiple component parts that interact in a nonlinear fashion; the results of complex systems are often greater than the sum of their parts.

### 4.2. Complex Whole Systems Approach

When developing, implementing and evaluating approaches aimed at reducing socioeconomic health inequalities, the complexity of the system should be considered. A complex system approach that follows the principles of systems thinking can contribute to this. Public Health England defines this type of approach as follows:
“A local whole systems approach responds to complexity through an ongoing, dynamic and flexible way of working. It enables local stakeholders, including communities, to come together, share an understanding of the reality of the challenge, consider how the local system is operating and where the best opportunities for change can be found. Stakeholders agree on actions and decide as a network how to work together in an integrated way to bring about sustainable, long term systems change”[81].


A systematic review of whole systems approaches to obesity and other complex public health challenges shows some positive health outcomes [82]. Process evaluations of whole systems approaches showed successful features, including embedding within a broader policy context, sufficient financial support and resources, time to build relationships, good governance, local evaluation, trust and capacity and full engagement of relevant partners and the community.

In order to develop, implement and evaluate a whole systems approach aimed at reducing socioeconomic health inequalities, multiple tools can help identify the determinants and their interactions within the system. First, causal loop diagrams can help explore interactions in the system and find leverage points for change in that system by creating a visual image of aspects of the system and their interactions [83]. This visualization and discussion about this visualization, preferably between different stakeholders, can help find leverage points to break through feedback loops, which may lead to the greatest impact higher up in the system [84]. Research should focus on what can be done to bring about change higher up in the system, for example, changing the organizational structure or the goals and beliefs of policymakers. Second, the realist evaluation approach takes the complexity of reality into account and can also be used to identify interactions in a complex system. Instead of answering the question ‘does it work?’ during evaluation, Pawson and Tilley state that the more relevant and realistic question is ‘what works, how, in which conditions and for whom?’ [85]. Realist evaluation acknowledges that mechanisms (M) of an approach may only work in a particular context (C), and that the combination of both can lead to a particular outcome (O). Identifying and testing these CMO configurations are the main aims of conducting realist evaluation and can help to better understand the complex system. Third, social network analysis (SNA) can help optimize collaboration within and between different domains [86]. The SNA technique aims to describe and analyze patterns of social interaction between different entities, for example, people or organizations. In this way, stakeholders that affect the system of socioeconomic health inequalities can be identified and encouraged to contribute to behavior change in low-SEP groups [87]. Fourth, in the phase of developing actions that contribute to a feasible system (for example, a system that reduces health inequalities), the action scales model can be used [84]. The action scales model aims to help practitioners and policymakers understand why the system functions the way it does and to identify opportunities for change through action across four levels. The model describes four levels (i.e., events, structures, goals and beliefs) and pays attention to coherence and synergy within the system.

### 4.3. Participation of All Stakeholders in System Change

The system of health-related behaviors does not only consist of determinants that can be measured objectively, but also of experiences and perceptions of different stakeholders. Participation of all stakeholders, including the often difficult-to-reach people from low-SEP groups, to build the most complete understanding of the system is an important aspect of complex systems thinking [88,89]. Moreover, the complex system of the person, environment, ball and their interactions, can vary between specific groups: the determinants and their interactions within the system might work differently for people from low-SEP groups in comparison with the general population. For these reasons, participation of people from these low-SEP groups specifically is important because they know best what is appropriate for them in stimulating healthy behavior. They can provide insight to other stakeholders into what motivates them, what their talents are and what hinders or encourages them in daily life. Consequently, the design and implementation of policy and interventions can be through co-creation aligned specifically with these aspects of people from this low-SEP group. Whether creating causal loop diagrams, identifying CMO configurations, social network analysis and brainstorming about actions on different levels of the action scale model, participation of all stakeholders and co-creation when using these tools leads to more suitable outcomes. A recent study by Mackenbach [90] noted that relative inequalities in European countries uniformly increased because all morbidity and mortality rates go down in low and high-SEP groups. However, some countries did see their absolute inequalities decrease. The author believed that what contributed to this was ensuring that effective interventions have sufficient reach in low-SEP groups, which can be done through participation [90]. With people from low-SEP groups actively involved in the development, implementation and evaluation of interventions and approaches, the interventions and approaches can be in line with the living environment, living conditions, skills, wishes and needs of the low-SEP groups and lead to more effectiveness [91].

Not only people from low-SEP groups should participate in systems thinking: other stakeholders within the system, e.g., policymakers, local professionals and managers ‘higher’ up in the system, should participate as well. As early as 2007, Burns [92] noted in his book on Systemic Action Research (SAR) methodology that changing complex settings requires a full system change involving all the relevant stakeholders because complex issues cannot be isolated from the wider system of which they are a part. Moreover, there is often a ‘gap’ between people from the low-SEP groups and policymakers because the latter often design policy measures without knowing what it is like to live in the circumstances of someone with a low SEP. Collaboration and co-creation between these two groups must, therefore, take place at all levels. One way to do this is by deploying and training ‘experts by experience’ who are able to share knowledge and experiences of low-SEP groups at a policy level [93]. According to Burns [94], SAR offers a ‘learning architecture’ where all relevant stakeholders are required to acknowledge and address the complexity. According to Burns, the following key elements are required for this system: support for the change on all levels, communication, bridging actions and the distribution of leadership within the system.

## 5. Conclusions

This study aimed to increase insight into why socioeconomic health inequalities are persistent and to build the way forward for improved approaches. We conclude that socioeconomic health inequalities might be persistent because policymakers, researchers and other professionals working on reducing health inequalities have not acknowledged (sufficiently) the complexity of systems of determinants at multiple levels and their dynamic interactions. It seems as if systems thinking is the way forward and this is increasingly acknowledged, but still much is unknown. There are three answers to the question of what, according to us, the way forward should be for policy, research and practice aimed at reducing socioeconomic health inequalities. First, it is important to acknowledge complex systems thinking while taking the interaction between individual, environment and behavior change into account. Second, it is important to investigate and address mechanisms that specifically influence the system for low-SEP groups because determinants in the system and their interactions can work differently for those groups. Third, participation of all stakeholders and sharing of knowledge and experiences of people from low-SEP groups on all levels is important in systems thinking. We do not know what the specific system in a city, neighborhood, school, family or individual looks like. For every situation, the ball, slope and person are different. Moreover, it is important to include stakeholders that can influence different aspects of the system.

Methodologies such as realist evaluation [85,95], social network analysis [87,96] and causal loop diagrams [97] can be used for the development and implementation of complex interventions and approaches aimed at reducing health inequalities that potentially have more impact than current approaches. Participation of all relevant stakeholders (including people from low-SEP groups) when using those methodologies and co-creation can help investigate and address this complex issue. We recommend using the action scale model to identify opportunities for change through action across the different levels presented in the model (i.e., events, structures, goals and beliefs) [85]. Moreover, it is important to monitor and evaluate the process and effects of development and implementation of whole system approaches. The ENCOMPASS framework that is specifically developed for evaluating approaches in complex adaptive systems could, for example, be used [98]. Research on the application of the mentioned methodologies and frameworks in different contexts could help to generate more insight into what works in systems thinking and whole system approaches. We hope that the overview of theories and methodologies provided in this paper can help to develop, implement and evaluate improved approaches and with that reduce socioeconomic health inequalities.

## Figures and Tables

**Figure 1 ijerph-19-08384-f001:**
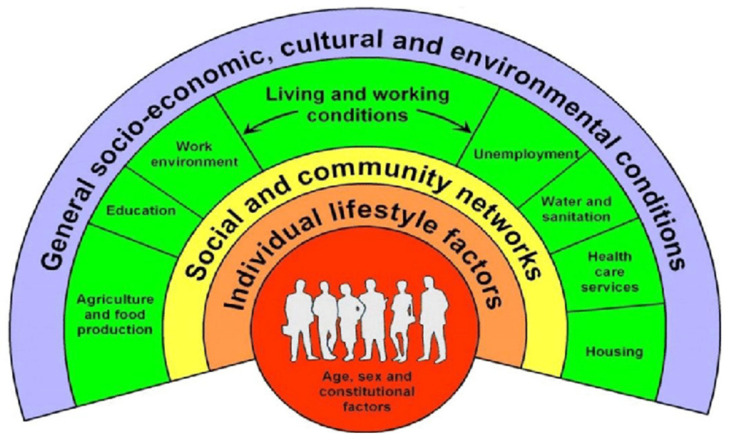
The main determinants of health. Reproduced with permission from Dahlgren, G. and Whitehead, M., European strategies for tackling social inequities in health; published by WHO, 2006.

**Figure 2 ijerph-19-08384-f002:**
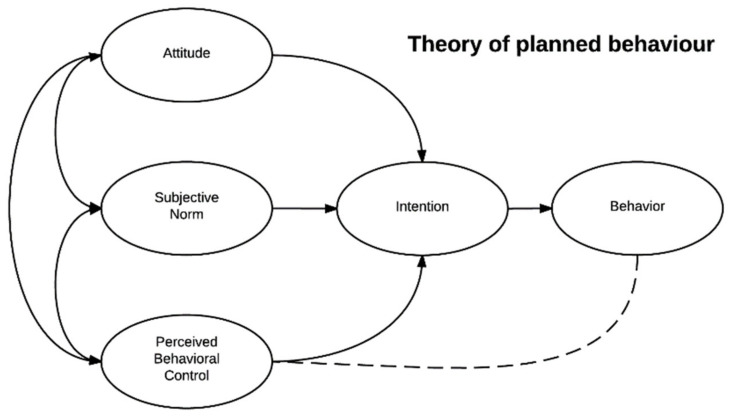
The theory of planned behavior. Reproduced from Ajzen, I., From Intentions to Actions: A Theory of Planned Behavior; published by Springer, 1985.

**Figure 3 ijerph-19-08384-f003:**
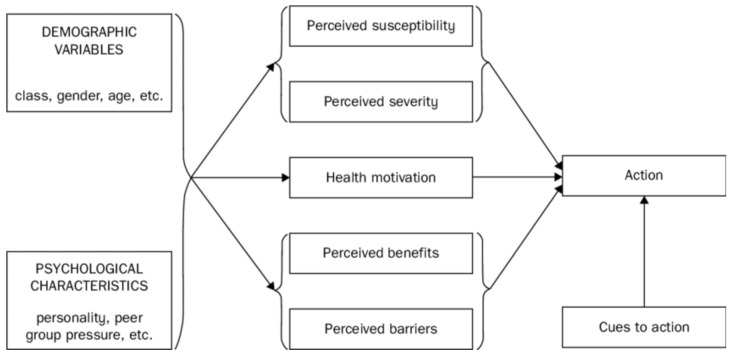
The health belief model. Reproduced from Rosenstock, I.M., Historical origins of the health belief model; published by Health Education Monographs, 1974.

**Figure 4 ijerph-19-08384-f004:**
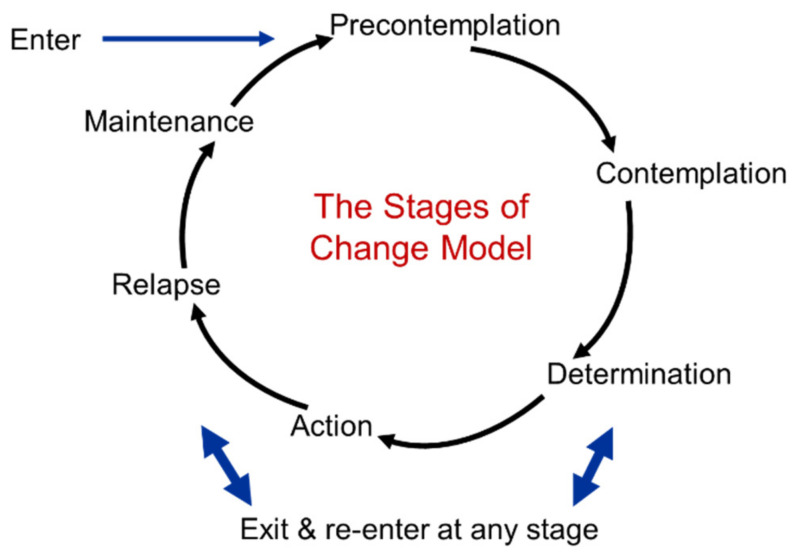
The stages of change model. Reproduced from Prochaska & Di Clemente, The transtheoretical model of health behavior change; published by The American Journal of Health Promotion, 1997.

**Figure 5 ijerph-19-08384-f005:**
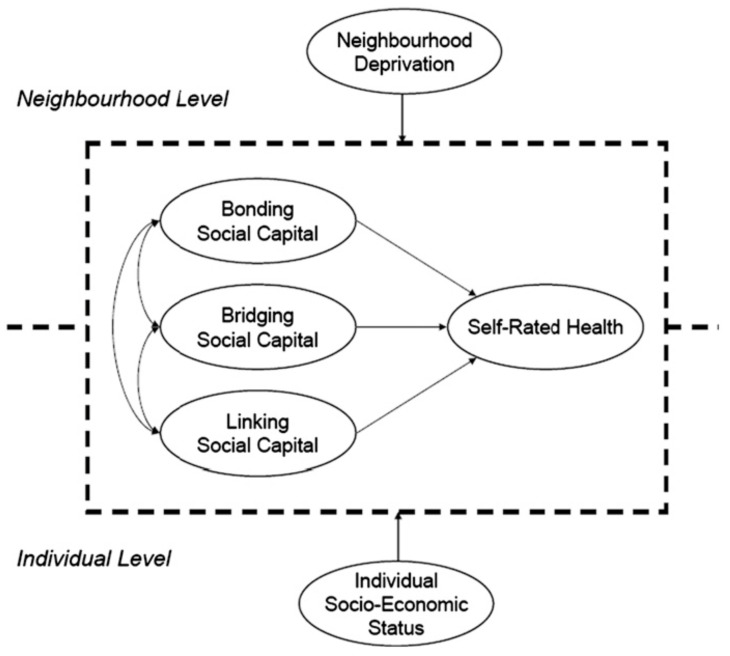
Social capital as determinants of health. Reproduced with permission from Israel, B.A., Social networks and health status: Linking theory, research, and practice; Published by Patient Counselling and Health Education, 1982.

**Figure 6 ijerph-19-08384-f006:**
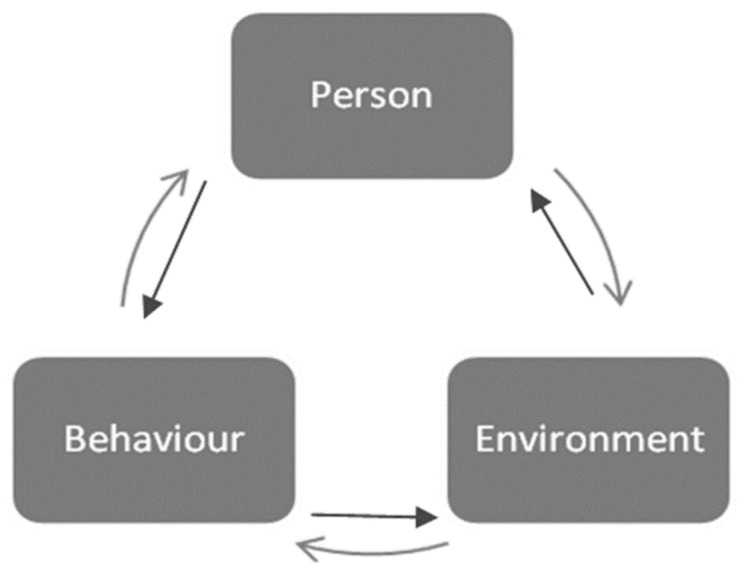
Structuration theory. Reproduced from Oppong, S., ‘Between Bandura and Giddens: Structuration Theory in Social Psychological Research?’; published by *Psychological Thought*, 2014 [56].

**Figure 7 ijerph-19-08384-f007:**
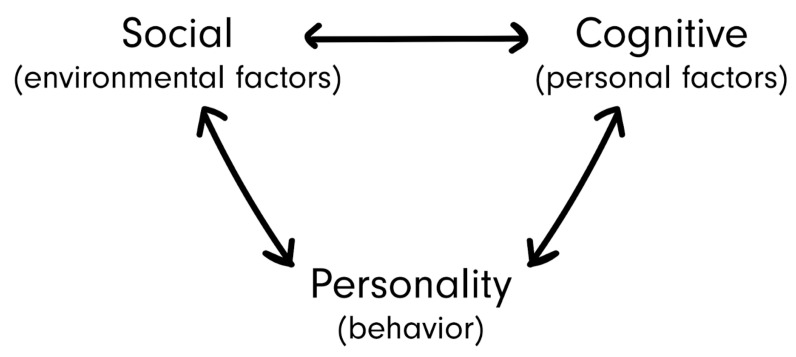
Social cognitive theory. Reproduced from Bandura, A., Human agency in social cognitive theory; published by American Psychologist, 1989.

**Figure 8 ijerph-19-08384-f008:**
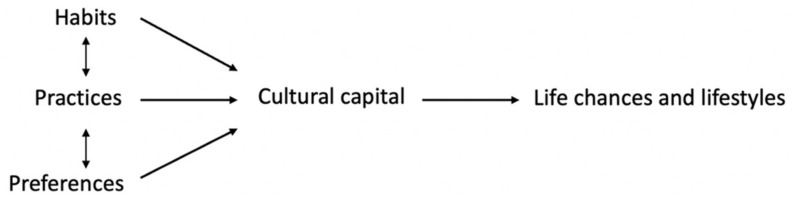
Cultural capital theory.

**Figure 9 ijerph-19-08384-f009:**
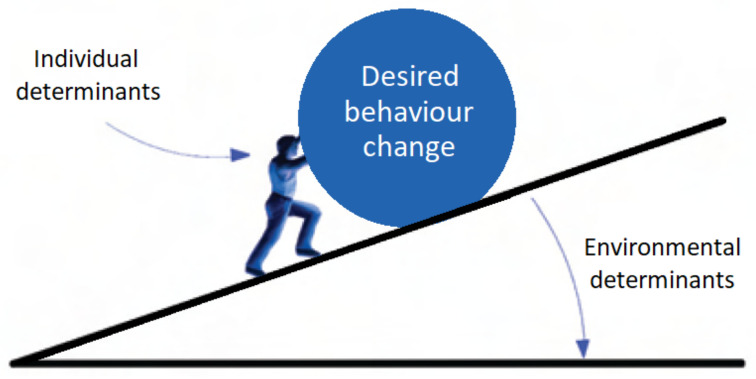
The complex system of the person, ball and slope.

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
