# Peer review of "A Theoretical Perspective on Why Socioeconomic Health Inequalities Are Persistent: Building the Case for an Effective Approach"

_ijerph, 2022, doi:10.3390/ijerph19148384_

Round 1

Reviewer 1 Report

The manuscript ‘A Theoretical Perspective on why Socioeconomic Health Inequalities are Persistent in the Netherlands: Building the Case for an Effective Approach’ sheds significant light to the existing research. However, major revisions are needed.

General comments:

1.      the title of the manuscript refers to the Netherlands and health inequalities in this country, but the manuscript lacks a description of the situation in this subject area;

2.      the structure of the manuscript is confusing - in the first part of the manuscript there are no subchapters dedicated to the description of particular theories as it is the case in the second and third parts of the manuscript. Instead, there is a subsection that describes the limitations of the theories mentioned above;

3.      the structure of the different parts of the manuscript should be unified;

4.      there is a lack of presentation of the theoretical framework of the study (for example graphical framework of the study);

5.      there is a lack of explanation on what basis the various theories relating to health behaviour/health behaviour change/health determinants were selected and used in this study;

Detailed comments:

1.      Each of the theories (models) described in the manuscript should be clearly presented for example in a diagram including/emphasizing elements that influence the emergence or persistence of health inequalities/ contribute to the reduction of socioeconomic health inequalities.

2.      the theoretical framework of the study should be presented, taking into account all described theories (models) relating to the emergence or reduction of social inequalities in health;

3.      It is worth presenting each of the theory or groups of theories described in the paricular parts of manuscript, including their applicability for addressing social inequalities in health and public health measures to reduce them;

4.      ‘Bronfenbrenner and Whitehead & Dahlgren’s models have been criticized for putting the individual at the center and not sufficiently highlighting the interaction between different determinants’. - I do not agree with this statement. The Dahlgren-Whitehead rainbow model remains one of the most effective illustrations of health determinants and has had widespread impact in research on health inequality. Characteristics shown in the center of the model are non-modifiable health determinants (those that are not able to be changed or controlled), but the individual lifestyle factors represents the group of the modifiable determinants of health;

5.      giving examples of activities/interventions based on the cited theories would greatly enrich the value of the work.

Reviewer 2 Report

The article is a free literature review, without specifying the type of review or meta-analysis. The article also does not describe the method of selecting and reviewing literature, nor is there a description of the method of analyzing theory and scientific evidence. Literature is more than 5 years old (over 40 items). The method of literature search must be accurately described.

Reviewer 3 Report

Comments on “A Theoretical Perspective on why Socioeconomic Health Inequalities are Persistent in the Netherlands: Building the Case for an Effective Approach

General Comments:

1.    This is a very well written and well structured presentation on a very relevant and timely topic.  It will certainly a very good addition to the investigations being undertaken on health equity.

2.    I have a few suggestions that I believe would strengthen the piece. The most important is revising the title.  As it stands it is more a reflection of the reason for undertaking the study. To interest the reader in the study I suggest a more succinct title.  Here is my suggestion. “An analytical framework to assess reasons for socio-economic inequalities for health outcomes”.  The fact that the Netherlands situation prompted the examination is not critical to the argument. The present is good for the introduction of the paper but not to attraction readers.  The presentation is much broader than the situation in the Netherlands.

3.    There are places in the argument that would be strengthened by examples.  See below.

Specific Comments

1.    Regarding  “keywords”, “theoretical perspective “is not useful.  Key words allow the reader to focus on specific topics.  This key word is very broad.

2.    P.2  Paragraph 2  the focus on “equity effect of health related behaviour” needs to be expanded.  Presenting an example would help.

3.     P. 3 “theoretical models show which determinants interact at different levels but do not show…”  Again an example would be most helpful.

4.    4 P.8 Figure 2.  In the text, it calls this figure, Figure 6.2.  Please correct the label.

Round 2

Reviewer 1 Report

The authors’ response letter and the revised manuscript have clearly addressed the points from the reviewer. Revised title is appropriate for the content of the article. Diagrams illustrating the various theories of behavior change and subsection titles are very helpful to reader.

Therefore, I am satisfied with the revision made on the manuscript and would like to recommend the manuscript for publication.